# Leveraging Invariant Principle for Heterophilic Graph Structure Distribution Shifts

## Abstract

Heterophilic Graph Neural Networks (HGNNs) have shown promising results for semi-supervised learning tasks on graphs. Notably, most real-world heterophilic graphs are composed of a mixture of nodes with different neighbor patterns, exhibiting local node-level homophilic and heterophilic structures. However, existing works are only devoted to designing better unified HGNN backbones for node classification tasks on heterophilic and homophilic graph benchmarks simultaneously, and their analyses of HGNN performance concerning nodes are only based on the determined data distribution without exploring the effect caused by the difference of structural pattern between training and testing nodes. How to learn invariant node representations on heterophilic graphs to handle this structure difference or distribution shifts remains unexplored. In this paper, we first discuss the limitations of previous graph-based invariant learning methods in addressing the heterophilic graph structure distribution shifts from the perspective of data augmentation. Then, we propose **HEI**, a framework capable of generating invariant node representations through incorporating **H**eterophily information, node's estimated neighbor pattern, to infer latent **E**nvironments without augmentation, which are then used for **I**nvariant prediction. We provide detailed theoretical guarantees to clarify the reasonability of HEI. Extensive experiments on various benchmarks and backbones can also demonstrate the effectiveness and robustness of our method compared with existing state-of-the-art baselines. Our codes can be accessed through HEI.

## Keywords

Graph Representation Learning, Node Classification, Invariant Learning, Distribution Shifts, Heterophily and Homophily

**ACM Reference Format:**
Anonymous Author(s). 2018. Leveraging Invariant Principle for Heterophilic Graph Structure Distribution Shifts. In *Proceedings of Make sure to enter the correct conference title from your rights confirmation emai (Conference acronym 'XX)*. ACM, New York, NY, USA, 14 pages. https://doi.org/XXXXXXX.XXXXXXX

## 1 Introduction

Graph Neural Networks (GNNs) have emerged as prominent approaches for learning graph-structured representations through the aggregation mechanism that effectively combines feature information from neighboring nodes [41]. Previous GNNs primarily dealt

with *homophilic graphs*, where connected nodes tend to share similar features and labels [42]. However, growing empirical evidence suggests that these GNNs' performance significantly deteriorates when dealing with *heterophilic graphs*, where most nodes connect with others from different classes, even worse than the traditional neural networks [19]. An appealing way to address this issue is to tailor the heterophily property to GNNs, extending the range of neighborhood aggregation and reorganizing architecture [41], known as the heterophilic GNNs (HGNNs).

*Heterophilic Graph Structure distribution Shift (HGSS): A novel data distribution shift perspective to reconsider existing HGNNs works.* Despite promising, most previous HGNNs assume the nodes share the determined data distribution [18, 19], we argue that there is data distribution disparity among nodes with different neighbor patterns. As illustrated in Figure 1(a1), heterophilic graphs are composed of a mixture of nodes that exhibit local homophilic and heterophilic structures, *i.e*, the nodes have different neighbor patterns [41]. The node's neighbor pattern can be measured by node homophily, representing homophily level by comparing the label between the node and its neighbors. Here, we identify their varying neighbor patterns between train and test nodes as the Heterophilic Graph Structure distribution Shift (Figure 1(a2)). This kind of shift was neglected by previous works but truly affected GNN's performance. As shown in Figure 1(a3), we visualize the HGSS between training and testing nodes on the Squirrel dataset. Compared with test nodes, the train nodes are more prone to be categorized into groups with high homophily, which may yield a test performance degradation. More statistical results on other heterophilic graph datasets can be shown in Figure 5. Notably, though some recent work [27] also discusses homophilic and heterophilic structural patterns, until now they haven't provided a clear technique solution for this problem. Compared with traditional HGNN works that focus on backbone designs, it's extremely urgent to seek solutions from a data distribution perspective to address the HGSS issue.

*Existing graph-based invariant learning methods perform badly for HGSS due to the augmentation-based environment construction strategy.* In the context of general distribution shifts, the technique of invariant learning [30] is increasingly recognized for its efficacy in mitigating these shifts. The foundational approach involves learning node representations to facilitate invariant predictor learning across various constructed environments (Figure 1(b1)), adhering to the Risk Extrapolation (REx) principle [9, 24, 37]. Unfortunately, previous graph-based invariant learning methods may not effectively address the HGSS issue, primarily due to explicit environments that may be ineffective for invariant learning. As illustrated in Figure 1(c1), within HGSS settings, altering the original structure does not consistently affect the node's neighbor patterns. In essence, obtaining optimal and varied environments pertinent to neighbor patterns is challenging. Our observation (Figure 1(c2)) reveals that EERM [37], a pioneering invariant learning approach utilizing environment augmentation to tackle graph distribution shifts in node-level

Figure 1: (a) illustrates the heterophilic graph structure distribution shifts (HGSS), where the figure and histogram show the HGSS and neighbor pattern (measured by node homophily) difference between train and test nodes on the Squirrel dataset; (b) displays the comparison of different environment construction strategies between previous invariant learning works and ours from augmentation; (c) shows that the environment construction of previous methods may be ineffective in addressing the HGSS due to the unchanged neighbor pattern distribution. The experimental results between traditional and graph-based invariant learning methods can support our analysis and verify the superiority of our proposed HEI.

tasks, does not perform well under HGSS settings. At times, its enhancements are less effective than simply employing the original V-Rex [16], which involves randomly distributing the train nodes across various environmental groups. We attribute this phenomenon to the irrational environment construction. According to our analysis, EERM is essentially a node environment-augmented version of V-Rex, *i.e.*, the disparity in their performance is solely influenced by the differing strategies in environmental construction. Besides, from the perspective of theory assumption, V-Rex is initially employed to aid model training by calculating the variance of risks introduced by different environments as a form of regularization. The significant improvements by V-Rex also reveal that the nodes of a single input heterophilic graph may reside in distinct environments, considering the variation in neighbor patterns, thus contradicting EERM's prior assumption that all nodes in a graph share the same environment [37]. Based on this insight, our goal is to break away from previous explicit environment augmentation to learn the latent environment partition, which empowers the invariant learning to address the HGSS better.

*HEI: Heterophily-Guided Environment Inference for Invariant Learning.* Recent studies explore the effect of prior knowledge on the environment partition [20, 33] and subsequently strengthen the importance of the environment inference and extrapolation for model generalization [36, 39]. Therefore, our initial step should be to quantify the nodes' neighbor pattern properties related to the HGSS, which are central to the issue at hand. Consequently, a critical question emerges: During the training phase, how can we identify an appropriate metric to estimate the node's neighbor pattern and leverage it to deduce latent environments to manage this HGSS issue? As previously mentioned, node homophily can assess

the node's neighbor patterns [19]. Unfortunately, this requires the actual labels of the node and its neighbors, rendering it inapplicable during the training stage due to the potential unlabeled status of neighbor nodes. To cope with this problem, several evaluation metrics pertinent to nodes' neighbor patterns, including local similarity [7], post-aggregation similarity [26], and SimRank [22], have been introduced. These metrics aim to facilitate node representation learning on heterophilic graphs during the training phase. But these studies primarily concentrate on employing these metrics to help select proper neighbors for improved HGNN architectures, while we aim to introduce a novel invariant learning framework-agnostic backbones to separate the spurious features from selected neighbors, tackling the structure distribution shifts. Therefore, we propose HEI, a framework capable of generating invariant node representations through incorporating heterophily information to infer latent environments, as shown in Figure 1 (b2), which are then used for downstream invariant prediction, under heterophilic graph structure distribution shifts. Extensive experiments on various backbones and benchmarks can verify the effectiveness of our proposed method in addressing this neglected HGSS issue.

**Our Contributions**: (i) We highlight an important yet often neglected form of heterophilic graph structure distribution shift, which is orthogonal to most HGNN works that focus on backbone designs; (ii) We propose HEI, a novel graph-based invariant learning framework to tackle the HGSS issue. Unlike previous efforts, our method emphasizes leveraging a node's inherent heterophily information to deduce latent environments without augmentation, thereby significantly improving the generalization and performance of HGNNs; (iii) We demonstrate the effectiveness of HEI on several benchmarks and backbones compared with existing methods.

## 2 Preliminaries

**Notations**. Given an input graph $G = (V, X, A)$, we denote $V \in \{v_1, ..., v_N\}$ as the nodes set, $X \in R^{N \times D}$ as node features and $A \in \{0, 1\}^{N \times N}$ as an adjacency matrix representing whether the nodes connect, where the $N$ and $D$ denote the number of nodes and features, respectively. The node labels can be defined as $Y \in \{0, 1\}^{N \times C}$, where C represents the number of classes. For each node $v$, we use $A_v$ and $X_v$ to represent its adjacency matrix and node feature.

**Problem Formulation**. We first provide the formulation for the general optimized object of node-level OOD (Out of Distribution) problem on graphs, then reclarify the formulation of previous works to help distinguish our work in the next section. From the perspective of data generation, we can get train data $(G_{train}, Y_{train})$ from train distribution $p(\mathbf{G}, \mathbf{Y})|e = e$, the model should handle the test data $(G_{test}, Y_{test})$ from a different distribution $p(\mathbf{G}, \mathbf{Y})|\mathbf{e} = e')$, varying in different environments $\mathbf{e}$. Thus, the optimized object of node-level OOD problem on graphs can be formulated as follows:

$$\min_{\omega, \Phi} \max_{e \in \mathcal{E}} \mathbb{E}_G \frac{1}{N} \sum_{v \in V} \mathbb{E}_{y \sim p(y|A_v = A_v, X_v = X_v, e=e)} l(f_\omega(f_\Phi(A_v, X_v)), y_v) \quad (1)$$

where the $\mathcal{E}$ represents the support of environments $e$, the $f_\omega$ and $f_\Phi$ refer to GNN's classifier and feature extractor respectively and $l(\cdot)$ is a loss function (e.g. cross entropy). The Eq 1 aims to learn a robust model that minimizes loss across environments as much as possible. Only in this way, can the trained model be likely to adapt to unknown target test distribution well. However, environmental labels for nodes are usually unavailable during the training stage, which inspires many works to seek methods to make use of environmental information to help model training.

**Previous Graph-based Invariant Learning.** To approximate the optimized object of Eq. 1, previous works [9, 24, 37] mainly construct diverse environments by adopting the masking strategy as Figure 1(b1). Thus, we conclude previous works from the masking strategy ($Mask_\eta(\cdot)$ parameterized with $\eta$). Given an input single graph, we can obtain K augmented graphs as Eq. 2, where each graph corresponds to an environment. The K is a pre-defined number of training environments and $X^m$, $A^m$, and $V^m$ are corresponding mask versions of feature, adjacency matrix, and node sets.

$$G^{e=k} = Mask_\eta^{e=k}(G) = (X^m, A^m, V^m)_{e=k}, k = 1, 2..., K \quad (2)$$

Then, assisted by these augmented graphs with environment label, the GNN $f(\cdot)$ parameterized by $(\omega, \Phi)$ can be trained considering environmental information. We can define the ERM (Empirical Risk Minimization) loss in the $k$-th environment as the Eq.3, which only calculates the loss on the corresponding augmented $G^{e=k}$.

$$R_{e=k}(\omega, \Phi) = \frac{1}{N} \sum_{v \in V^m} l(f_\omega(f_\Phi(A_v, X_v)), y_v) \quad (3)$$

Following the principle of Variance Risk Extrapolation (V-Rex) to reduce the risks from different environments, the final training framework can be defined as Eq. 4. Where the $\lambda$ controls the effect between reducing the average risk and promoting equality of risks.

$$\min_{\omega, \Phi} \max_{\eta} \ L(\Phi, \omega, \eta) =$$
$$\sum_{k=1}^{K} R_{e=k}(\omega, \Phi) + \lambda Var(R_{e=1}(\omega, \Phi), \cdots, R_{e=K}(\omega, \Phi)) \quad (4)$$

The maximization means that we should optimize the masking strategy (parameter $\eta$) to construct sufficient and diverse environments, while the minimization aims to reduce the training loss for the model (parameter $\omega$ and $\Phi$).

**Discussions.** Exactly, previous graph-based invariant learning methods introduce extra augmented graphs to construct nodes' environments while our work only infers nodes' environments on a single input graph. Specifically, there exists a latent assumption for previous works that nodes on a single graph belong to the same environment so we need to construct diverse environments by data augmentation. This assumption arises from the insight that nodes on an input graph come from the same *outer domain-related environments* (e.g. Financial graphs or Molecular graphs) [37]. But considering the message-passing mechanism on heterophilic graphs (the ideal aggregation target should be nodes with the same label), the nodes should exist exactly in *inner structure-related environments*. To cope with this issue, as shown in Figure 1(c1), directly utilizing *data augmentation may be ineffective in changing the node's neighbor pattern distribution* to construct diverse environments for invariant prediction. At the same time, the neighbor pattern difference between train and test has verified that even on a single graph, the nodes may belong to different structure-related environments. These simultaneously inspire us to directly infer node environments on a single graph assisted by the node's neighbor pattern, rather than constructing environments from different augmented graphs, for addressing heterophilic graph structure distribution shift.

## 3 Methodology

In this section, we present the details of the proposed HEI. Firstly, on heterophilic graphs, we verify that the similarity can serve as a neighbor pattern indicator and then review existing similarity-based metrics to estimate the neighbor patterns during training stages. Then, we elaborate the framework to jointly learn environment partition and invariant node representation on heterophilic graphs without augmentation, assisted by the estimated neighbor patterns. Finally, we clarify the overall training process of the algorithm and discuss its complexity. Moreover, we provide a detailed theoretical analysis in Appendix A.2 to clarify the details of HEI.

### 3.1 Neighbor Patterns Estimation

The node homophily is commonly used to evaluate the node's neighbor patterns, representing the node's structural pattern distribution [19]. Unfortunately, it needs the true labels of the node and its neighbors, which means they can not be used in the training stage because the neighbor nodes may be just the test nodes without labels for node classification tasks when given an input graph. To cope with it, we aim to utilize the similarity between node features to estimate the node's neighbor pattern.

**Similarity: An Indicator of Neighbor Patterns.** Previous works have shown there exists some relationship between similarity and homophily from the experimental analysis [7], it can not be guaranteed to work well without a theory foundation. Thus, we further investigate its effectiveness from the node cluster view and verify

**Figure 2: Illustrations of our framework HEI. (a) The neighbor pattern for each train node can be estimated by similarity first and then used for inferring environments without augmentation; (b) Based on the train nodes belonging to different inferred environments, we can train a set of environment-independent GNN classifiers with the shared encoder compared with the base GNN. The shared encoder outputs the representations of nodes in each environment and then forwards them to the base GNN classifier and the environment-independent classifier respectively. By calculating the loss gap between these two different classifiers, an invariance penalty is introduced to improve model generalization.**

the similarity between nodes can be exploited to approximate the neighbor pattern without the involvement of label information.

For simplicity, we take K-Means as the cluster algorithm. For two nodes $v$ and $u$, let $v$ belong to the cluster centroid $c_1$ and denote the square of the distance between $v$ and $u$ as $\delta = \|u - v\|^2$, we can get $c_1 = \arg\min \|v - c_i\|^2$, where $c_i$ represent the $i$-th cluster centroid. Then the distance between $u$ and cluster centroid $c_1$ can be acquired as the Eq. 5. Exactly, the neighbor pattern describes the label relationship between the node and its neighbors. From the Eq 5, we can find the smaller $\delta$, the more likely the $v$ and $u$ belong to the same cluster and own the same label. Therefore, the similarity between nodes can be exploited to serve as a neighbor pattern indicator without using label information.

$$
\begin{aligned}
\|u - c_1\|^2 &= \|(u - v) + (v - c_1)\|^2 \\
&= \|(u - v)\|^2 + 2\|u - v\| \|v - c_1\| + \|v - c_1\|^2 \\
&= \delta + 2\sqrt{\delta} \|v - c_1\| + \|v - c_1\|^2 \\
&= \left(\|v - c_1\| + \sqrt{\delta}\right)^2 \geq \delta
\end{aligned}
\tag{5}
$$

**Existing Similarity-based Metrics.** Existing similarity-based metrics on heterophilic graphs can be shown as Eq.6.

$$
Similarity(u, v) = \begin{cases} \text{Sim}(X_v, X_u) & \text{Local Sim} \\ \text{Sim}(\hat{A}_v X_v, \hat{A}_u X_u) & \text{Agg Sim} \\ \frac{c}{|NS(u)||NS(v)|} \sum_{\substack{u' \in NS(u) \\ v' \in NS(v)}} \text{Sim}(X_{u'}, X_{v'}) & \text{SimRank} \end{cases}
\tag{6}
$$

where the $c \in (0, 1)$ is a decay factor empirically set to 0.6, the $NS(v)$ denotes $v$'s neighbor set including the nodes connected to $v$ ,the $\hat{A}_v$ denotes the aggregation operation on the node $v$ and the $Sim$ denote the similarity calculation between two objects.. We can observe that the local similarity (Local Sim [7]) and post-aggregation similarity (Agg-Sim [26]) respectively calculate the similarity of the original and post-aggregation embedding between two nodes. In contrast, the SimRank [22] calculates the similarity between their respective neighbor nodes.

**Estimated Node's Neighbor Pattern.** Thus, as Eq.7, we can obtain the estimated neighbor patterns $z_v$ for the node $v$ during the training stage by averaging the node's similarity with neighbors.

$$
z_v = \frac{1}{|NS(v)|} \sum_{u \in NS(v)} Similarity(u, v)
\tag{7}
$$

Notably, we further strengthen our object of using similarity metrics is indeed different from previous HGNN works that utilize the similarity metrics[7, 22, 26] to design backbones. From the perspective of causal analysis shown in Figure 4, when given the neighbors, we aim to separate and weaken the effect of spurious features from full neighbor features by utilizing the estimated neighbor pattern to infer the node's environment for invariant prediction. However, previous HGNN works mainly aim to help the node select proper neighbors and then directly utilize full neighbor features as aggregation targets for better HGNN backbone designs. Our work is exactly *orthogonal* to previous HGNN works.

## 3.2 HEI: Heterophily-Guided Environment Inference for Invariant Learning

We aim to utilize the estimated neighbor patterns $Z \in R^{G_z}$, which represent the node's heterophily information, as an auxiliary instrument to jointly learn nodes' environment partition and invariant node representation without augmentation. Similar techniques can be also shown in [3, 20] for image classification tasks. Specifically, assisted by the estimated neighbor patterns for nodes, we can train an environment classifier $\rho(\cdot) : R^{G_z} \to R^K$ that softly assigns the train nodes to $K$ environments. The $K$ is a pre-defined number, $\rho$ is a two-layer MLP and the $\rho^{(k)}(\cdot)$ is denoted as the $k$-th entry of $\rho(\cdot)$, with $\rho(Z) \in [0,1]^K$ and $\sum_k \rho^{(k)}(Z) = 1$. Denote the ERM loss calculated on all train nodes as $R(\omega, \Phi)$. Then, as shown in Eq. 8, the ERM loss in the $k$-th inferred environment can be defined as $R_{\rho^{(k)}}(\omega, \Phi)$, which only calculates the loss on the nodes belonging to the $k$-th environment.

**Overal Framework:** Based on the above analysis, the training framework of HEI can be defined as follows:

$$R_{\rho^{(k)}}(\omega, \Phi) = \frac{1}{N} \sum_{v \in V} \rho^{(k)}(z_v) l\left(f_\omega\left(f_\Phi\left(A_v, X_v\right), y_v\right)\right) \quad (8)$$

$$\min_{\omega, \Phi} \max_{\rho, \{\omega_1, \cdots, \omega_K\}} L(\Phi, \omega, \omega_1, \cdots, \omega_K, \rho) =$$

$$R(\omega, \Phi) + \lambda \underbrace{\sum_{k=1}^{K}\left[R_{\rho^{(k)}}(\omega, \Phi) - R_{\rho^{(k)}}(\omega_k, \Phi)\right]}_{\text{invariance penalty}} \quad (9)$$

Compared with previous graph-based invarinat learning methods shown in Eq. 3 and Eq. 4, our framework mainly differs in the maximization process. Thus, we clarify the effectiveness and reasonability of our framework from two aspects: (i) The invariance penalty learning that introduces a set of environment-dependent GNN classifiers $\{f_{\omega_k}\}_{k=1}^{K}$, which are only trained on the data belonging to the inferred environments; (ii) The adaptive environment construction through optimizing the environmental classifier $\rho(\cdot)$.

**Invariance Penalty Learning.** As shown by Eq.1, the ideal GNN classifier $f_\omega$ is expected to be optimal across all environments. After the environment classifier $\rho^{(k)}(\cdot)$ assigns the train nodes into k inferred environments, we can adopt the following criterion to check if $f_\omega$ is already optimal in all inferred environments: Take the $k$-th environment as an example, we can additionally train an environment-dependent classifier $f_{\omega_k}$ on the train nodes belonging to the $k$-th environment. If $f_{\omega_k}$ achieves a smaller loss, it indicates that $f_\omega$ is not optimal in this environment. Moreover, we can further train a set of classifiers $\{f_{\omega_k}\}_{k=1}^{K}$, each one with a respective individual environment, to assess whether $f_\omega$ is simultaneously optimal in all environments. Notably, all these classifiers share the same encoder $f_\Phi$, if $f_\Phi$ extracts spurious features that are unstable across the inferred environments, $R_{\rho^{(k)}}(\omega, \Phi)$ will be larger than $R_{\rho^{(k)}}(\omega_k, \Phi)$, resulting in a non-zero invariance penalty, influencing model optimization towards achieving optimality across all environments. In other words, as long as the encoder extracts the invariant feature, the GNN classifier $f_\omega$ and its related environment-dependent classifier $\{f_{\omega_k}\}_{k=1}^{K}$ will have the same prediction across

---

**Algorithm 1** HEI: Heterophily-Guided Environment Inference for Invariant Learning

---

1: **Require:** Graph data $G$ and label $Y$; Environment classifier $\rho$; GNN feature encoder $f_\Phi$; GNN classifier $f_\omega$; Number of training environments: $K$; a set of Environment-independent GNN classifiers $\{f_{\omega_k}\}_{k=1}^{K}$;
2: Estimate neighbor pattern $z$ for each node by Eq. 7;
3: **while** Not converged or maximum epochs not reached **do**
4:     Divide the nodes into $K$ environments by $\rho(k)(z)$ and obtain corresponding split graphs $\{G_e = k\}_{k=1}^{K}$;
5:     **for** $k = 1, \cdots, K$ **do**
6:         Calculate the GNN's loss on the train nodes belonging to the $k$-th environment, $R_{\rho^{(k)}}(\omega, \Phi)$, via Eq. 8;
7:         Train an additional environment-independent GNN classifier $f_{\omega_k}$ with the shared GNN feature encoder $f_\Phi$ on the train nodes belonging to the $k$-th inferred environment, calculate its loss $R_{\rho^{(k)}}(\omega_k, \Phi)$;
8:     **end for**
9:     Calculate invariance penalty and the total loss via Eq. 9;
10:     Update $\rho$ via maximizing the invariance penalty;
11:     Update $f_\Phi, f_\omega$ via minimizing the total loss;
12: **end while**

---

different environments.

**Adaptive Environment Construction.** As shown in Figure 1(c), the effectiveness of previous methods is only influenced by environmental construction strategy. A natural question arises: What is the ideal environment partition for invariant learning to deal with the HGSS? We investigate it from the optimization of environment classifier $\rho(\cdot)$. Specifically, a good environment partition should construct environments where the spurious features exhibit instability, incurring a large penalty if $f_\Phi$ extracts spurious features. In this case, we should maximize the invariance penalty to optimize the partition function $\rho(\cdot)$ to generate better environments, which is also consistent with the proposed strategy. Though previous works [9, 24, 37] also adopt the maximization process to construct diverse environments, they just focus on directly optimizing the masking strategy to get augmentation graphs. During the optimization process, these methods lack guidance brought by auxiliary information $Z$ related to environments, ideal or effective environments are often unavailable in this case. That's why we propose to introduce the environment classifier to infer environments without augmentation, assisted by the $Z$. Exactly, to make sure the guidance of $Z$ has a positive impact on constructing diverse and effective environments for the invariant node representation learning, there are also two conditions for $Z$ from the causal perspective. We will further clarify it in Appendix A.2.

### 3.3 Training Process and Complexity Analysis

**Training Process:** As shown by Algorithm 1: Given a heterophilic graph input, we first estimate the neighbor patterns for each train node by Eq. 7. Then, based on Eq. 8 and Eq. 9, we aim to learn environment partition and invariant node representation, assisted by the estimated neighbor patterns through a min-max alternative

optimization. Specifically, maximizing the invariance penalty is devoted to optimizing the environmental classifier to construct as diverse environments as possible to enlarge the loss gap between the base GNN feature encoder and additionally introduced GNN feature encoders. In contrast, the minimization of total loss aims to promote the base GNN to learn invariant representation agnostic neighbor patterns to address the HGSS issues.

**Complexity Analysis:** Given a graph with $N$ nodes, the average degree is $d$. GNN with $l$ layers calculate embeddings in time and space $O(Nld^2)$. HEI assigns N nodes into $k$ inferred environments $N_{e=1} + \cdots + N_{e=k} = N$ and executes $k + 1$ classifier computations, where the $k$ corresponds to the $k$ environment-independent classifiers, and 1 refers to the basic GNN classifier. Denote $N'$ as the average number of nodes belonging to an inferred environment, the overall time complexity is $O(Nld^2 + kN'ld^2)$, which is linear to the scale of the graph. More detailed efficiency studies compared with previous methods can be shown in Experiments.

## 4 Experiments

In this section, we investigate the effectiveness of HEI to answer the following questions.

- **RQ1:** Does HEI outperform state-of-art methods to address the HGSS issue?
- **RQ2:** How robust is the proposed method? Can HEI solve the problem that exists in severe distribution shifts?
- **RQ3:** How do different similarity-based metrics influence the neighbor pattern estimation, so as to further influence the effect of HEI?
- **RQ4:** What is the sensitivity of HEI concerning the predefined number of training environments?
- **RQ5:** How efficient is the proposed HEI compared with previous methods?

### 4.1 Experimental Setup

**Datasets**. We adopt six commonly used heterophilic graph datasets (chameleon, Squirrel, Actor, Penn94, arxiv-year, and twitch-gamer) and three homophilic graph datasets (Cora, CiteSeer and PubMed) to verify the effectiveness of HEI [19, 28]. To make sure the evaluation is stable and reasonable, we utilize the filtered versions of existing datasets to avoid data leakage [29]. Notably, considering that we should further split the test datasets to construct different evaluation settings. Those excessive small-scale heterophilic graph datasets, such as Texa, Cornell, and Wisconsin [28], are not fit and chosen for evaluation due to their unstable outcomes. Moreover, considering the nodes on homophilic graphs means there exists a mere structure difference between train and test nodes. We just provide experiments and discussions in the Appendix A.3.

**Settings.** Based on previous dataset splits, we construct two different settings to evaluate the effectiveness and robustness of HEI: **(i) Standard Settings:** We sort the test nodes based on their nodes' homophily values and acquire the median. The part that is higher than the median is defined as the High Hom Test, while the rest is defined as the Low Hom Test. The model is trained on the previous train dataset and evaluated on more fine-grained test groups; **(ii)**

**Simulation Settings where exists severe distribution shifts.:** We sort and split the train and test nodes simultaneously adopting the same strategy of (i). The model is trained on the Low/High Hom Train and evaluated on the High/Low Hom Test.

**Backbones.** To further verify our framework is orthogonal to previous HGNN works that focus on backbone designs, we adapt HEI to two existing SOTA and scalable backbones with different foundations, LINKX (MLP-based) [19] and GloGNN++ (GNN-based) [18]. In this way, our improvements can be attributed to the design that deals with the neglected heterophilic structure distribution shifts.

**Baselines**. Denote the results of the backbone itself as ERM. Our comparable baselines can be categorized into: (i) Reweight-based methods considering structure information: Renode [4] and StruRW-Mixup [23]; (ii) Invariant Learning methods involving environment inference for node-level distribution shift: SRGNN [43], EERM [37], BAGNN [9], FLOOD [24], CaNet [36] and IENE [39] ; (iii) Prototype-based methods for structural distribution shift on the special domain(e.g. graph anomaly detection): GDN [11]. Notably, though we can utilize estimated neighbor patterns as auxiliary information to infer environments related to HGSS, the true environment label is still unavailable. So we don't compare with those traditional invariant learning methods that rely on the explicit environment labels, e.g. IRM [2], V-Rex [16] and GroupDRO [31].

### 4.2 Experimental Results and Analysis

**Handling Distribution Shifts under Standard Settings (RQ1).** We first evaluate the effectiveness of HEI under standard settings, where we follow the previous dataset splits and further evaluate the model on more fine-grained test groups with low and high homophily, respectively. The results can be shown in Table 1 and Table 2. We have the following observations.

On the one hand, *the impact brought by the HGSS is still apparent though we adopted the existing SOTA HGNN backbones.* As shown by the results of ERM in Table 1 and Table 2, for most datasets, there are significant performance gaps between the High Hom Test and Low Hom Test, ranging from 5 to 30 scores. These results further verify the necessity to seek methods from the perspective of data distribution rather than backbone designs to deal with this problem.

On the other hand, *HEI can outperform previous methods in most circumstances.* Specifically, compared with invariant learning methods, though HEI does not augment the training environments, utilizing the estimated neighbor patterns to directly infer latent environments still benefits invariant prediction and improves model generalization on different test distributions related to homophily. In contrast, directly adopting a reweight strategy (Renode and StruRW) or evaluating the difference between the training domain and target domain (SRGNN) without environment augmentation can't acquire superior results than invariant learning methods. This is because these methods need accurate domain knowledge or structure information in advance to help model training. However, for the HGSS issue, the nodes' environments on heterophily graphs are unknown and difficult to split into the invariant and spurious domains, like the GOOD dataset [12] which has clear domain and distribution splits. Simultaneously, the neighbor pattern distribution represents more fine-grained label relationships between

 

**Table 1: Performance comparison on small-scall heterophilic graph datasets under Standard Settings. The reported scores denote the classification accuracy (%) and error bar (±) over 10 trials. We highlight the best score on each dataset in bold and the second score with underline.**

| Backbones | Methods | Chamelon-filter | | | Squirrel-filter | | | Actor | | |
|---|---|---|---|---|---|---|---|---|---|---|
| | | Full Test | High Hom Test | Low Hom Test | Full Test | High Hom Test | Low Hom Test | Full Test | High Hom Test | Low Hom Test |
| LINKX | ERM | 23.74 ± 3.27 | 29.24 ± 3.87 | 16.81 ± 3.21 | 37.11 ± 1.68 | 45.11 ± 1.68 | 28.17 ± 1.54 | 35.83 ± 1.40 | 38.21 ± 1.76 | 33.42 ± 1.94 |
| | ReNode | 24.14 ± 3.51 | 29.64 ± 3.42 | 17.24 ± 3.25 | 37.91 ± 1.84 | 45.81 ± 1.58 | 28.97 ± 1.24 | 28.62 ± 1.79 | 33.92 ± 3.67 | 23.25 ± 1.54 |
| | StruRW-Mixup | 24.19 ± 3.81 | 29.84 ± 3.58 | 17.46 ± 3.27 | 37.98 ± 1.57 | 45.88 ± 1.81 | 29.27 ± 1.24 | 28.62 ± 1.79 | 33.92 ± 3.67 | 23.25 ± 1.54 |
| | SRGNN | 24.42 ± 3.57 | 29.94 ± 3.59 | 17.96 ± 3.12 | 38.11 ± 1.67 | 45.98 ± 1.82 | 29.57 ± 1.23 | 29.57 ± 1.81 | 34.45 ± 3.51 | 24.67 ± 1.34 |
| | EERM | 24.54 ± 3.61 | 30.12 ± 3.71 | 18.22 ± 3.42 | 38.57 ± 1.77 | 46.24 ± 1.82 | 29.57 ± 1.23 | 29.62 ± 1.94 | 36.50 ± 3.21 | 22.66 ± 1.26 |
| | BAGNN | 24.64 ± 3.71 | 30.17 ± 3.61 | 18.29 ± 3.54 | 38.61 ± 1.72 | 46.14 ± 1.93 | 29.87 ± 1.15 | 36.10 ± 2.01 | 38.46 ± 2.16 | 33.49 ± 1.86 |
| | FLOOD | 24.64 ± 3.62 | 30.15 ± 3.62 | 18.25 ± 3.82 | 38.59 ± 1.81 | 46.33 ± 1.52 | 29.64 ± 1.13 | 36.40 ± 2.31 | 38.72 ± 2.17 | 33.98 ± 1.76 |
| | CaNet | 24.71 ± 3.55 | 30.28 ± 3.28 | 18.45 ± 3.78 | 38.75 ± 1.85 | 46.45 ± 1.55 | 29.74 ± 1.34 | 36.58 ± 2.31 | 38.92 ± 2.44 | 34.12 ± 1.48 |
| | IENE | 24.74 ± 3.58 | 30.30 ± 3.62 | 18.85 ± 3.82 | 38.81 ± 1.86 | 46.43 ± 1.59 | 29.84 ± 1.75 | 36.65 ± 2.31 | 38.99 ± 2.35 | 34.13 ± 1.96 |
| | GDN | 24.65 ± 3.52 | 30.31 ± 3.62 | 18.15 ± 3.82 | 38.89 ± 1.81 | 46.58 ± 1.57 | 29.94 ± 1.87 | 36.38 ± 2.25 | 38.52 ± 2.17 | 34.08 ± 1.76 |
| | **HEI(Ours)** | **25.78 ± 2.23** | **31.35 ± 2.56** | **20.21 ± 4.21** | **40.92 ± 1.31** | **48.82 ± 2.88** | **31.21 ± 2.79** | **37.41 ± 1.17** | **39.31 ± 1.45** | **35.42 ± 1.54** |
| GloGNN++ | ERM | 25.90 ± 3.58 | 31.40 ± 3.63 | 20.10 ± 2.51 | 35.11 ± 1.24 | 41.25 ± 1.88 | 27.61 ± 1.54 | 37.70± 1.40 | 40.96 ± 1.52 | 34.37 ± 1.61 |
| | ReNode | 25.99 ± 3.88 | 31.50 ± 3.83 | 20.20 ± 2.32 | 35.54 ± 1.12 | 41.54 ± 1.78 | 28.21 ± 1.59 | 29.82 ± 1.79 | 35.42 ± 2.75 | 25.24 ± 2.18 |
| | StruRW-Mixup | 26.12 ± 3.58 | 31.70 ± 3.53 | 20.30 ± 2.12 | 35.64 ± 1.18 | 41.70 ± 1.58 | 28.31 ± 1.56 | 29.87 ± 1.81 | 35.45 ± 2.79 | 25.34 ± 2.51 |
| | SRGNN | 26.72 ± 3.68 | 32.20 ± 3.43 | 21.00 ± 2.52 | 36.34 ± 1.28 | 42.30 ± 1.58 | 28.75 ± 1.54 | 30.87 ± 1.79 | 35.62 ± 2.75 | 25.64 ± 2.54 |
| | EERM | 26.99 ± 3.58 | 32.51 ± 3.73 | 21.22 ± 2.41 | 36.54 ± 1.38 | 42.70 ± 1.48 | 29.45 ± 1.55 | 32.75 ± 2.41 | 39.34 ± 3.21 | 26.98 ± 2.87 |
| | BAGNN | 27.12 ± 3.48 | 32.61 ± 3.83 | 21.42 ± 2.71 | 36.64 ± 1.52 | 42.89 ± 1.49 | 29.55 ± 1.61 | 38.05 ± 1.29 | 41.26 ± 1.52 | 34.87 ± 1.87 |
| | FLOOD | 27.17 ± 3.58 | 32.81 ± 3.63 | 21.82 ± 2.61 | 36.84 ± 1.42 | 42.97 ± 1.58 | 29.85 ± 1.57 | 38.35 ± 1.59 | 41.54 ± 1.34 | 34.99 ± 2.11 |
| | CaNet | 27.37 ± 3.27 | 32.99 ± 3.87 | 22.08 ± 2.51 | 36.99 ± 1.51 | 43.17 ± 1.66 | 30.15 ± 1.44 | 38.37 ± 1.61 | 41.58 ± 1.32 | 35.01± 2.10 |
| | IENE | 27.75 ± 3.31 | 33.35 ± 3.97 | 22.62 ± 2.49 | 37.15 ± 1.66 | 43.37 ± 1.66 | 30.38 ± 1.64 | 38.38 ± 1.66 | 41.58 ± 1.37 | 35.08 ± 2.12 |
| | GDN | 27.21 ± 3.68 | 32.91 ± 3.52 | 21.84 ± 2.21 | 36.66 ± 1.42 | 43.12 ± 1.57 | 29.65 ± 1.58 | 38.39 ± 1.69 | 41.59 ± 1.74 | 35.12 ± 2.51 |
| | **HEI(Ours)** | **29.31 ± 3.68** | **34.35 ± 3.52** | **24.25± 2.71** | **39.42 ± 1.51** | **45.19 ± 1.57** | **31.45 ± 1.68** | **39.41 ± 1.51** | **42.25 ± 1.59** | **36.12 ± 1.85** |

**Table 2: Performance comparison on large-scall heterophilic graph datasets under Standard Settings. The reported scores denote the classification accuracy (%) and error bar (±) over 5 trials. We highlight the best score on each dataset in bold and the second score with underline.**

| Backbones | Methods | Penn94 | | | arxiv-year | | | twitch-gamer | | |
|---|---|---|---|---|---|---|---|---|---|---|
| | | Full Test | High Hom Test | Low Hom Test | Full Test | High Hom Test | Low Hom Test | Full Test | High Hom Test | Low Hom Test |
| LINKX | ERM | 84.67 ± 0.50 | 87.95 ± 0.73 | 81.07 ± 0.50 | 54.44 ± 0.20 | 64.74 ± 0.42 | 48.39 ± 0.62 | 66.02 ± 0.20 | 85.47 ± 0.66 | 46.38 ± 0.67 |
| | ReNode | 84.91 ± 0.41 | 88.02 ± 0.79 | 81.53 ± 0.88 | 54.46 ± 0.21 | 64.80 ± 0.37 | 48.37 ± 0.55 | 66.13 ± 0.14 | 84.25 ± 0.48 | 47.84 ± 0.43 |
| | StruRW-Mixup | 84.96 ± 0.43 | 88.11 ± 0.56 | 81.91 ± 0.78 | 54.35 ± 0.21 | 64.78 ± 0.32 | 48.31 ± 0.81 | 66.10 ± 0.12 | 84.29 ± 0.47 | 47.89 ± 0.58 |
| | SRGNN | 84.98 ± 0.37 | 87.92 ± 0.79 | 81.83 ± 0.78 | 54.42 ± 0.20 | 64.80 ± 0.37 | 48.38 ± 0.54 | 66.15 ± 0.09 | 84.45 ± 0.48 | 48.01 ± 0.43 |
| | EERM | 85.01 ± 0.55 | 87.81 ± 0.79 | 82.08 ± 0.71 | 54.82 ± 0.32 | **68.06 ± 0.61** | 46.46 ± 0.61 | 66.02 ± 0.18 | 83.27 ± 0.40 | 48.39 ± 0.34 |
| | BAGNN | 85.02 ± 0.37 | 88.21 ± 0.68 | 82.02 ± 0.59 | 54.65 ± 0.30 | 66.46 ± 0.57 | 47.56 ± 0.58 | 66.17 ± 0.12 | 83.77 ± 0.40 | 48.56 ± 0.59 |
| | FLOOD | 85.07 ± 0.32 | 88.25 ± 0.59 | 82.11 ± 0.61 | 54.77 ± 0.29 | 66.81 ± 0.59 | 47.88 ± 0.56 | 66.18 ± 0.14 | 83.85 ± 0.42 | 48.71 ± 0.61 |
| | CaNet | 85.10 ± 0.28 | 88.33 ± 0.54 | 82.15 ± 0.61 | 54.78 ± 0.29 | 66.88 ± 0.57 | 47.91 ± 0.66 | 66.20 ± 0.15 | 83.95 ± 0.42 | 48.78 ± 0.61 |
| | IENE | 85.15 ± 0.32 | 88.25 ± 0.59 | 82.20 ± 0.60 | 54.80 ± 0.33 | 66.85 ± 0.61 | 48.01 ± 0.59 | 66.21 ± 0.12 | 84.45 ± 0.47 | 48.81 ± 0.61 |
| | GDN | 85.19 ± 0.37 | 88.31 ± 0.68 | 82.32 ± 0.59 | 54.75 ± 0.30 | 66.96 ± 0.61 | 47.86 ± 0.58 | 66.12 ± 0.12 | 83.77 ± 0.40 | 48.56 ± 0.59 |
| | **HEI(Ours)** | **86.22 ± 0.28** | **89.24 ± 0.28** | **83.22 ± 0.59** | **56.05 ± 0.22** | 66.53 ± 0.41 | **49.33 ± 0.32** | **66.79 ± 0.14** | **85.53 ± 0.25** | **49.21 ± 0.57** |
| GloGNN++ | ERM | 85.81 ± 0.43 | 89.51 ± 0.82 | 81.75 ± 0.58 | 54.72 ± 0.27 | 65.78 ± 0.41 | 48.12 ± 0.72 | 66.29 ± 0.20 | 84.25 ± 1.06 | 48.13 ± 1.06 |
| | Renode | 85.81 ± 0.42 | 89.53 ± 0.81 | 81.75 ± 0.57 | 54.76 ± 0.25 | 65.91 ± 0.54 | 48.12 ± 0.75 | 66.32 ± 0.16 | 84.01 ± 0.56 | 48.44 ± 0.67 |
| | StruRW-Mixup | 85.92 ± 0.37 | 89.83 ± 0.81 | 81.81 ± 0.47 | 54.81 ± 0.35 | 65.98 ± 0.64 | 48.52 ± 0.65 | 66.29 ± 0.15 | 84.21 ± 0.56 | 48.54 ± 0.67 |
| | SRGNN | 85.89 ± 0.42 | 89.63 ± 0.81 | 82.01 ± 0.57 | 54.69 ± 0.25 | 65.87 ± 0.44 | 48.39 ± 0.85 | 66.25 ± 0.16 | 84.31 ± 0.56 | 48.34 ± 0.57 |
| | EERM | 85.86 ± 0.33 | 89.41 ± 0.74 | 81.97 ± 0.50 | 53.11 ± 0.19 | 61.03 ± 0.54 | 48.54 ± 0.43 | 66.20 ± 0.30 | 83.97 ± 1.18 | 48.25 ± 0.96 |
| | BAGNN | 85.95 ± 0.27 | 89.61 ± 0.74 | 81.92 ± 0.40 | 54.81 ± 0.17 | 66.07 ± 0.44 | 48.39 ± 0.33 | 66.22 ± 0.25 | 83.77 ± 0.85 | 48.64 ± 0.59 |
| | FLOOD | 85.99 ± 0.31 | 89.64 ± 0.67 | 82.05 ± 0.51 | 54.89 ± 0.22 | 66.22 ± 0.42 | 48.55 ± 0.31 | 66.24 ± 0.22 | 83.81 ± 0.79 | 48.50 ± 0.54 |
| | CaNet | 86.02 ± 0.35 | 89.69 ± 0.70 | 82.11 ± 0.57 | 54.91 ± 0.33 | 66.28 ± 0.42 | 48.68 ± 0.33 | 66.28 ± 0.27 | 83.89 ± 0.78 | 48.62 ± 0.59 |
| | IENE | 86.11 ± 0.30 | 89.64 ± 0.68 | 82.18 ± 0.47 | 54.86 ± 0.19 | 66.21 ± 0.44 | 48.65 ± 0.33 | 66.35 ± 0.25 | 84.11 ± 0.78 | 48.79 ± 0.44 |
| | GDN | 85.92 ± 0.41 | 89.53 ± 0.67 | 81.75 ± 0.51 | 54.76 ± 0.17 | 66.24 ± 0.44 | 48.15 ± 0.19 | 66.21 ± 0.27 | 83.78 ± 0.89 | 48.42 ± 0.51 |
| | **HEI(Ours)** | **87.18 ± 0.28** | **89.99 ± 0.65** | **83.59 ± 0.39** | **55.71 ± 0.24** | **66.29 ± 1.14** | **49.52 ± 0.75** | **66.99 ± 0.17** | **84.37 ± 0.68** | **50.40 ± 0.52** |

nodes and their neighbors, which means it's more complex and challenging compared to previous long-tail degree or label problems depending on directly counting class types and numbers of neighbors. Moreover, the great performance from GDN verifies the necessity of learning node representation close to its class prototype through regularization, mitigating the effect of neighbor patterns during aggregation. However, HEI can still outperform GDN due to the more fine-grained causal feature separation depending on the constructed environments related to the node's neighbor patterns, further verifying the effectiveness of HEI.

**Handling Distribution Shifts under Simulation Settings where exists severe distribution shifts (RQ2).** As shown in Figure 3 and Figure 6 in Appendix A.3, for each dataset, we mainly report results

under severe distribution shifts between training and testing nodes, which include $Train_{High}$ on $Test_{Low}$ and $Train_{Low}$ on $Test_{High}$. From the results, we can observe that HEI achieves better performance than other baselines apparently, with up to 5 ∼ 10 scores on average. In contrast, previous methods with environment augmentation achieved similar improvements. This is because all of them succeed in the augmentation-based environmental construction strategy on the ego-graph of train nodes to help the model adapt to diverse distribution, which may be ineffective under the HGSS scenarios, especially when there exists a huge structural pattern gap between train and test nodes. These results can further verify the robustness and effectiveness of the proposed HEI for handling the HGSS issues.

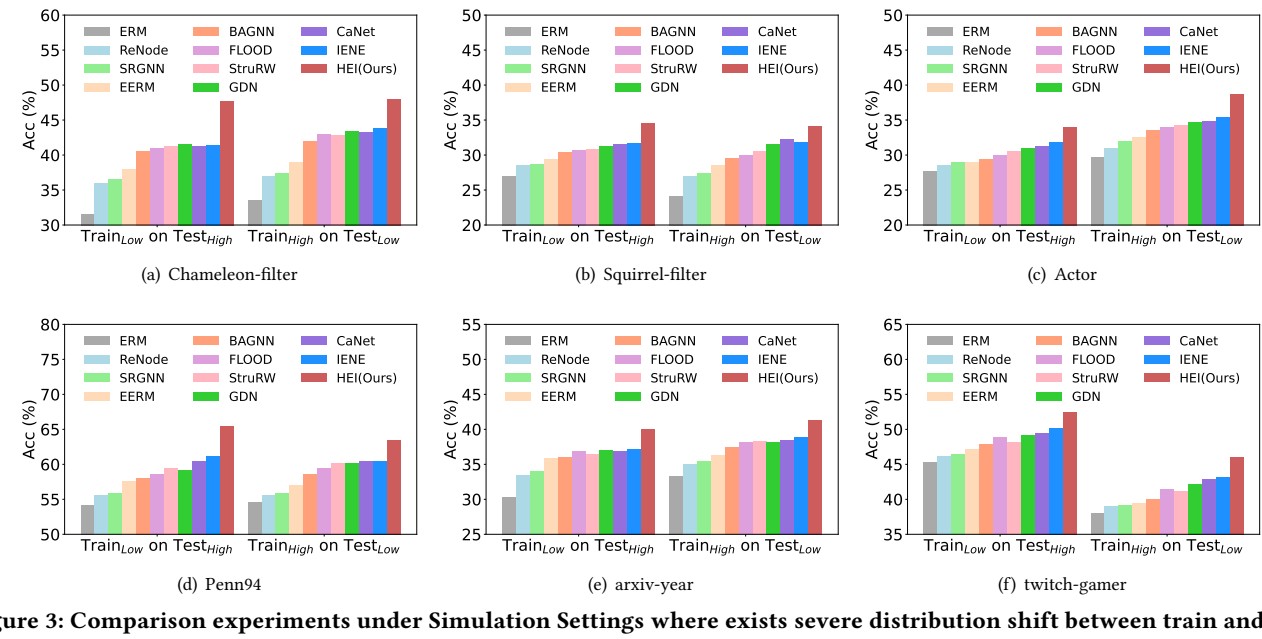

**Figure 3: Comparison experiments under Simulation Settings where exists severe distribution shift between train and test, including $Train_{High}$ on $Test_{Low}$ and $Train_{Low}$ on $Test_{High}$. We adopt the LINKX as the backbone there.**

**Table 3: Comparison experiments on small-scale datasets when we respectively adopt local similarity(Local Sim), post-aggregation similarity (Agg Sim), and SimRank as indicators to estimate nodes' neighbor patterns so as to infer latent environments under Standard settings.**

| Backbones | Methods | Chamelon-filter | | | Squirrel-filter | | | Actor | | |
|---|---|---|---|---|---|---|---|---|---|---|
| | | Full Test | High Hom Test | Low Hom Test | Full Test | High Hom Test | Low Hom Test | Full Test | High Hom Test | Low Hom Test |
| LINKX | HEI (Local Sim) | 24.91 ± 3.68 | 30.84 ± 3.52 | 18.95 ± 3.77 | 38.99 ± 1.85 | 46.81 ± 1.87 | 30.15 ± 1.95 | 36.84 ± 1.75 | 38.99 ± 2.44 | 34.51 ± 1.88 |
| | HEI (Agg Sim) | 25.11 ± 3.81 | 30.99 ± 3.92 | 19.17 ± 3.99 | 39.15 ± 1.99 | 47.15 ± 1.97 | 30.85 ± 2.11 | 36.87 ± 1.54 | 39.11 ± 2.48 | 34.68 ± 1.74 |
| | HEI (SimRank) | **25.78 ± 2.23** | **31.35 ± 2.56** | **20.21 ± 4.21** | **40.92 ± 1.31** | **48.82 ± 2.88** | **31.21 ± 2.79** | **37.41 ± 1.17** | **39.31 ± 1.45** | **35.42 ± 1.54** |
| GloGNN++ | HEI (Local Sim) | 27.78 ± 3.51 | 33.25 ± 3.41 | 22.51 ± 2.17 | 37.55 ± 1.55 | 44.11 ± 1.41 | 30.11 ± 1.28 | 38.57 ± 1.39 | 41.84 ± 1.44 | 35.41 ± 2.41 |
| | HEI (Agg Sim) | 28.24 ± 3.27 | 33.74 ± 3.11 | 22.94 ± 2.27 | 37.99 ± 1.25 | 44.54 ± 1.28 | 30.66 ± 1.51 | 38.81 ± 1.64 | 41.95 ± 1.55 | 35.81 ± 2.23 |
| | HEI (SimRank) | **29.31 ± 3.68** | **34.35 ± 3.52** | **24.25 ± 2.71** | **39.42 ± 1.51** | **45.19 ± 1.57** | **31.45 ± 1.68** | **39.41 ± 1.51** | **42.25 ± 1.59** | **36.12 ± 1.85** |

**The effect of different similarity metrics as neighbor pattern indicators for HEI (RQ3).** As depicted in Table 3 and Table 7 in the Appendix, we can draw two conclusions: (i) Applying any similarity-based metrics can outperform the previous SOTA strategy. This verifies the flexibility and effectiveness of HEI and helps distinguish HEI from previous HGNN works that also utilize the similarity for backbone designs; (ii) Applying SimRank to HEI can acquire consistently better performance than other metrics. This can be explained by previous HGNN backbone designs [7, 22, 26], which have verified that SimRank has a better ability to distinguish neighbors patterns compared with Local Sim and Agg-Sim, so as to design a better HGNN backbone, SIMGA[22]. Moreover, from the perspective of definitions as Eq. 6, the SimRank is specifically designed considering structural information, which is more related to our problem considering structure-related distribution shifts.

**Sensitivity Analysis of HEI concerning the pre-defined environmental numbers $K$ (RQ4).** As shown in Figure 7 in Appendix A.3, we vary the hyper-parameter environment numbers $k$ in Eq. 9 within the range of [2, 12], and keep all other configurations unchanged to explore its impact. We can observe that $K$ has a stable effect on the HEI, especially when $k \geq 6$, which can further verify its effectiveness in addressing HGSS issues.

**Efficiency Studies (RQ5).** As shown in Table 5 in Appendix A.3, we provide the time to train the model until converges that keep the stable accuracy score on the validation set. Experiments of these large-scale datasets are conducted on a single Tesla V100 GPU with 32G memory and use AdamW as the optimizer following [18]. From the results, we can conclude that the extra time cost can be acceptable compared with the backbone itself and other baselines.

## 5 Conclusion

In this paper, we emphasize an overlooked yet important variety of graph structure distribution shifts that exist on heterophilic graphs. We verify that previous node-level invariant learning solutions with environment augmentation are ineffective due to the irrationality of constructing environments. To mitigate the effect of this distribution shift, we propose HEI, a framework capable of generating invariant node representations by incorporating the estimated neighbor pattern information to infer latent environments without augmentation, which are then used for downstream invariant learning. Experiments on several benchmarks and backbones demonstrate the effectiveness of our method to cope with this graph structure distribution shift. Finally, we hope this study can draw attention to the structural distribution shift of heterophilic graphs.

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

# A Appendix

In this appendix, we provide the details omitted in the main text due to the page limit, offering additional experimental results, analyses, proofs, and discussions.

- A.1: We provide a detailed literature review related to our work and further distinguish our work with these works to clarify our contribution.
- A.2: We provide detailed theoretical analysis from the causal perspective to clarify the reasonability of HEI: Why we can utilize the estimated neighbor pattern to infer environments to address the HGSS issue?
- A.3: We provide the full experimental results on small and large-scale datasets adopting LINKX and GloGNN as backbones ( 4.2 of the main paper).
- A.4: We provide detailed implementation details to reproduce our experiments.

## A.1 Related Work

**Graph Neural Networks with Heterophily.** Existing strategies for mitigating graph heterophily issues can be categorized into two groups [41]: (i) Non-Local Neighbor Extension, aimed at identifying suitable neighbors through mixing High-order information [1, 14, 42] or discovering potential neighbors assisted by various similarity-based metrics [13–15, 22, 34, 35]; (ii) GNN Architecture Refinement, focusing on harnessing information derived from the identified neighbors, through selectively aggregating distinguishable and discriminative node representations, including adapting aggregation scheme [18, 25, 32], separating Ego-neighbor [25, 32, 42] and combining inter-layer [5, 10, 42].

However, these efforts share the common objective of designing better unified HGNN backbones for node classification tasks on heterophilic and homophilic graph benchmarks simultaneously. Moreover, we also found a recent arxiv paper called INPL [40] that coincides with our work. Though it also mentions the distribution shifts of neighborhood patterns, it still focuses on proposing an adaptive Neighborhood Propagation (ANP) module to *optimize the HGNN backbone or architecture without a thorough analysis of previous graph-based invariant-learning methods*. Different from these works, we instead consider from an identifiable neighbor pattern distribution perspective and propose a novel invariant learning framework that can be integrated with most HGNN backbones to further enhance their performance and generalization.

**Generalization on GNNs.** Many efforts have been devoted to exploring the generalization ability of GNNs. **(i) For graph-level tasks**, it assumes that every graph can be treated as an instance for prediction tasks [37]. Many works propose to identify invariant sub-graphs that decide the label Y and spurious sub-graphs related to environments, such as CIGA [8] , GIL [17], GREA [21], DIR [38] and GALA [6] (ii) **However, for node-level tasks that we focus on in this paper**, the nodes are interconnected in a graph as instances in a non-iid data generation way, it is not feasible to transfer graph-level strategies directly. To address this issue, EERM [37] proposes to regard the node's ego-graph with corresponding labels as instances and assume that all nodes in a graph often share the same environment, so it should construct different

environments by data augmentation, *e.g.*, DropEdge [30]. Based on these findings, BA-GNN [9], FLOOD [24] and IENE [39] inherit this assumption to improve model generalization. Apart from these environments-augmentation methods, the SR-GNN [43] and GDN [11] are two works that address distribution shifts on node-level tasks from the domain adaption and prototype learning perspectives respectively. Moreover, Renode [4] and StruRW-Mixup [23] are two reweight-based methods that explore the effect brought by the structure difference between nodes for node classification tasks. We also compare them in experiments.

Unlike these works, we highlight a special variety of structure-related distribution shifts for node classification tasks on heterophilic graphs and propose a novel invariant learning framework adapted to heterophilic graphs without dealing with graph augmentation to address this problem.

## A.2 Theoretical Analysis

To help understand our framework well, we first provide the comparison between our work and previous graph-based invarinat learning works as shown in Figure 4. Specifically, the definitions of random variables can be defined as follows: We define $\mathbf{G}$ as a random variable of the input graph, $\mathbf{A}$ as a random variable of node's neighbor information, $\mathbf{X}$ as a random variable of node's features, and $\mathbf{Y}$ as a random variable of node's label vectors. Both node features $\mathbf{X}$ and node neighbor information $\mathbf{A}$ consist of invariant predictive information that determines the label $\mathbf{Y}$ and the spurious information influenced by latent environments $\mathbf{e}$. In this case, we can denote $\mathbf{X} = \left[\mathbf{X}^I, \mathbf{X}^S\right]$ and $\mathbf{A} = \left[\mathbf{A}^I, \mathbf{A}^S\right]$.

Then, we provide a more detailed theoretical analysis of our framework from a casual perspective to identify invariant features.

**Casual conditions of $Z$.** From the casual perspective, some conditions are also needed for $Z$ to make sure our framework to address the heterophilic graph structure distribution shifts well [20]. Denote the $H(Y|X, A)$ as the expected loss of an optimal classifier over $(X, A,$ and $Y)$, we can clarify the reasonability of utilizing the estimated neighbor patterns as $Z$ auxiliary information to infer environments for invariant prediction based on the following two conditions.

CONDITION 1 (INVARIANCE PRESERVING CONDITION). *Given invariant feature $(X^I$ and $A^I)$ and any function $\rho(\cdot)$, it holds that*

$$H(Y|(X^I, A^I), \rho(Z)) = H(Y|(X^I, A^I)). \quad (10)$$

CONDITION 2 (NON-INVARIANCE DISTINGUISHING CONDITION). *For any feature $X^{Sk} \in X^S$ or $A^{Sk} \in A^S$ ,there exists a function $\rho(\cdot)$ and a constant $C > 0$ satisfy:*

$$H(Y|(X^{Sk}, A^{Sk})) - H(Y|(X^{Sk}, A^{Sk}), \rho(Z)) \geq C. \quad (11)$$

Condition 1 requires that invariant features $X^I$ and $A^I$ should keep invariant under any environment split obtained by $\rho(Z)$. Otherwise, if there exists a split where an invariant feature becomes non-invariant, then this feature would introduce a positive penalty as shown in Eq. 9 to further promote the learning of invariant node representation. Exactly, Condition 1 can be met only if $H(Y|(X^I, A^I), Z) = H(Y|(X^I, A^I))$, which means the auxiliary variable $Z$ should be d-separated by invariant feature $X^I$ and $A^I$. We provide a detailed proof in the appendix A.2. Exactly, the estimated

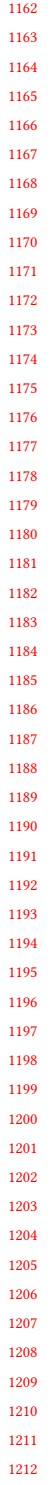

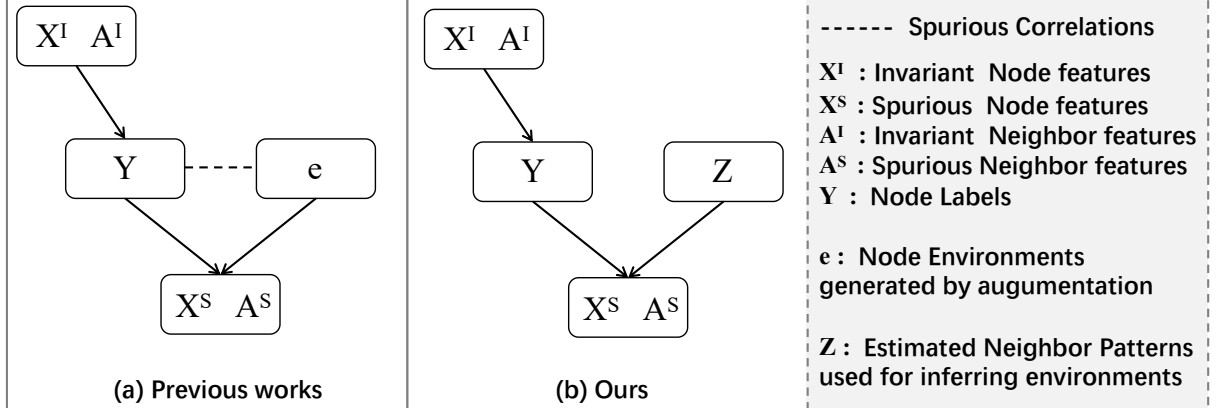

**Figure 4: Comparisons between our work and previous graph-based invariant learning works from the causal perspective. Notably, the basic HGNN directly aggregates the selected neighbors' full features without further separating like the above two types of invariant learning methods.**

neighbor pattern just describes the similarity between the node and its neighbors as Eq. 7, while the label $Y$ only has the direct causal relationship with $X^I$ and $A^I$ from the causal perspective. This means the Condition 1 can be met by adopting the estimated neighbor pattern as auxiliary information $Z$ to construct environments.

Condition 2 reveals that for each spurious feature $X^S$ and $A^S$, there exists at least one environment split where this feature demonstrates non-invariance within the split environment. If a spurious feature doesn't cause invariance penalties in all environment splits, it can't be distinguished from true invariant features. As shown in Figure 1(c2), the results of V-Rex are better than ERM, which means even randomly split environments with seeds can have a positive effect on making spurious features produce effective invariance penalty, further promoting the learning of invariant features. It's more likely to construct comparable or better environments than random seeds under the guidance of the estimated neighbor patterns $Z$. Thus, Condition 2 can also be guaranteed under our defined heterophilic graph structure distribution shit.

**Proof of Meeting Condition 1.** We show that for all $\rho(\cdot)$, if $H(Y|(X^I, A^I), Z) = H(Y|(X^I, A^I))$ holds, then there will exist that $H(Y|(X^I, A^I), \rho(Z)) = H(Y|(X^I, A^I))$.

PROOF. On one hand, because $\rho(Z)$ contains less information than $Z$, we have

$$H(Y|(X^I, A^I), \rho(Z)) \geq H(Y|(X^I, A^I), Z) = H(Y|(X^I, A^I)).$$

On the other hand, $(X^I, A^I)$ and $\rho(Z)$ contain more information than $(X^I, A^I)$, so we can get

$$H(Y|(X^I, A^I), \rho(Z)) \leq H(Y|(X^I, A^I)).$$

Thus, we conclude $H(Y|(X^I, A^I), \rho(Z)) = H(Y|(X^I, A^I))$.

**Assumptions and Theorem to Identify Invariant Features.** In particular, our assumption is consistent with previous invariant learning [20]. So we provide the previous version to support our framework, where $X = [X_v; X_s]$, where the $X_v$ refers to the invariant feature and the $X_s$ refers to the spurious feature.

ASSUMPTION 1. *For a given feature mask $\Phi$ and any constant $\epsilon > 0$, there exists $f \in F$ such that $E[l(f(\Phi(X)), Y)] \leq H(Y|\Phi(X)) + \epsilon$.*

ASSUMPTION 2. *If a feature violates the invariance constraint, adding another feature would not make the penalty vanish, i.e., there exists a constant $delta > 0$ so that for spurious feature $X_1 \subset X_s$ and any feature $X_2 \subset X$,*
$$H(Y|X_1, X_2) - H(Y|\rho(Z), X_1, X_2) \geq \delta (H(Y|X_1) - H(Y|\rho(Z), X_1)).$$

ASSUMPTION 3. *For any distinct features $X_1, X_2, H(Y|X_1, X_2) \leq H(Y|X_1) - \gamma$ with fixed $\gamma > 0$.*

Exactly, Assumption 1 is a common assumption that requires the function space $F$ be rich enough such that, given $\Phi$, there exists $f \in F$ that can fit $P(Y|\Phi(X))$ well. Assumption 2 aims to ensure a sufficient positive penalty if a spurious feature is included. Assumption 3 indicates that any feature contains some useful information w.r.t. $Y$, which cannot be explained by other features. Otherwise, we can simply remove such a feature, as it does not affect prediction.

The theorem to identify invariant features can be defined:

THEOREM 1. *Depending on the Assumptions 1-3 and Conditions 1-2, if $\epsilon < \frac{C\gamma\delta}{4\gamma + 2C\delta H(Y)}$ and $\lambda \in [\frac{H(Y)+1/2\delta C}{\delta C - 4\epsilon} - \frac{1}{2}, \frac{\gamma}{4\epsilon} - \frac{1}{2}]$, then we will have $\hat{L}(\Phi_v) < \hat{L}(\Phi)$ for all $\Phi \neq \Phi_v$, where $H(Y)$ denotes the entropy of $Y$.*

## A.3 More Experimental Results

We provide more experimental results to further show the effectiveness of the proposed HEI in addressing heterophilic graph structure distribution shifts.

**RQ2: Additional Experiments on Simulation Settings using GloGNN++ as backbone.** We also conduct experiments under severe distribution shifts using GloGNN++ as the backbone. As shown in Figure 6, our proposed method can acquire superior or comparable results than previous methods to handle graph structure distribution shifts, which further verifies the effectiveness and robustness of our design.

**Table 4: Statistics for our used heterophilic graph datasets.**

| Dataset | Chameleon | Squirrel | Actor | Penn94 | arXiv-year | twitch-gamer |
|---------|-----------|----------|-------|--------|------------|--------------|
| **Nodes** | 2277 | 5201 | 7600 | 41554 | 169343 | 168114 |
| **Edges** | 36101 | 216933 | 29926 | 1362229 | 1166243 | 6797557 |
| **Feat** | 2325 | 2089 | 931 | 5 | 128 | 7 |
| **Class** | 5 | 5 | 5 | 2 | 5 | 2 |
| **Edge hom.** | 0.23 | 0.22 | 0.22 | 0.47 | 0.222 | 0.545 |

**RQ3: Effect of different similarity matrices as neighbor pattern indicators for HEI.** We provide large-scale graph experiments as shown in Table 7 to clarify the details of HEI.

**RQ4: Sensitive analysis.** We provide the experimental results about RQ4 there as shown in Figure 7.

**RQ5: Efficiency Studies.** As shown in Table 5, referring to [18], we provide the time(seconds) to train the model until it converges which keeps the stable accuracy score on the validation set. From the results, we can conclude that the extra time cost can be acceptable compared with the backbone itself.

**Experiments on Homophilic Graph Datasets.** Considering the fact that in real-world settings, we can't know whether the input graph is homophilic or heterophilic in advance. Thus, we also provide comparison experiments and discussions for homophilic graphs. As shown in Table 6, from the results, we observe that our method can achieve consistent comparable performance to other baselines. But exactly, the improvements by these methods are all minor compared with the results of ERM. That's because the homophilic graph is not related to our settings. After all, homophilic graph datasets mean the neighbor pattern distribution between the train and test are nearly the same, which is not suitable to clarify our defined distribution shifts. The performance gap between the low home test and the high home test can support our analysis.

**Table 5: Efficiency studies of HEI, where the report scores show the time (seconds) to train the model until converge that keeps the stable accuracy score on the validation set. Referring to [18], we adopt the GloGNN++ as the backbone there.**

| Methods | Penn94 | arxiv-year | twitch-gamer |
|---------|--------|------------|--------------|
| ERM | 22.3 | 7.2 | 40.5 |
| Renode | 23.5 | 8.5 | 41.2 |
| SRGNN | 24.9 | 9.1 | 41.0 |
| EERM | 24.7 | 8.8 | 41.5 |
| BAGNN | 24.8 | 9.1 | 42.1 |
| FLOOD | 24.5 | 8.8 | 41.8 |
| StruRW | 23.8 | 9.8 | 41.5 |
| GDN | 24.7 | 9.6 | 42.2 |
| CaNet | 25.9 | 10.8 | 42.2 |
| IENE | 25.4 | 10.6 | 42.8 |
| HEI(Ours) | 26.8 | 11.5 | 44.9 |

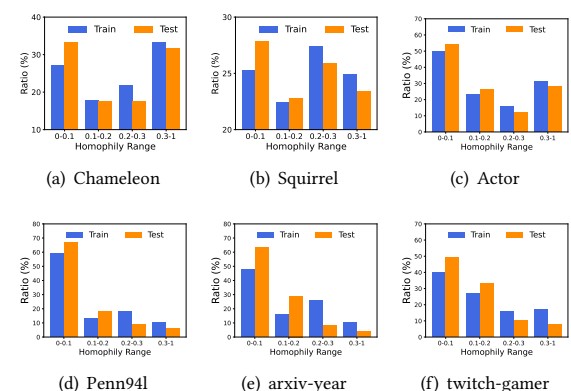

(a) Chameleon     (b) Squirrel     (c) Actor

(d) Penn94l     (e) arxiv-year     (f) twitch-gamer

**Figure 5: Statistic of homophily ratio for train and test nodes following previous dataset splits. The nodes are categorized into four groups according to the node-level homophily. Compared with test nodes, the train nodes are more prone to be categorized into groups with high homophily. In other words, in the range with high homophily(from 0.2 to 0.3 and from 0.3 to 1), the sub-train ratio in all train nodes is higher than the sub-test in all test nodes. But in the range with low homophily, there exists a contrary phenomenon.**

## A.4 Implementation Details

We provide detailed implementation details for our experiments.

**ERM**, It corresponds to the results of the backbone itself [18, 19] under our constructed settings.

**SRGNN**[43], Shift-Robust GNN is a framework inspired by domain adaption, which means it needs prior knowledge from the target domain. Specifically, it strives to adapt a biased sample of labeled nodes to more closely conform to the distributional characteristics present in an IID sample of the graph. In our experiments, we utilize the estimated neighbor distribution information to evaluate the distribution of the source domain and target domain. Apart from this, we entirely follow this work to address graph structure distribution shifts on heterophilic graphs.

**Renode**[4], in our experiments, it is an extension of the original Renode, which is a model-agnostic training weight schedule mechanism to cope with topology-imbalance problems for semi-supervised node classification. Specifically, they devise a cosine annealing mechanism for the training node weights based on their Totoro values. The Totoro values involve topology information and can also integrate with quantity-imbalanced methods as the paper shown. Therefore, this method can also be seen as a baseline which can copy with agnostic distribution shifts. We adopt the same reweight strategy as the paper stated, and the Totoro values can also describe the neighbor pattern to a certain extent.

**EERM**[37], Explore-to-Extrapolate Risk Minimization (EERM) is an invariant learning approach that facilitates graph neural networks to leverage invariance principles for prediction. As stated in the paper, EERM resorts to multiple context explorers that are adversarially trained to construct diverse environments with augmentation. Then, following the principle of variance of risks, it

**Table 6: Performance comparison on homophilic graph datasets under Standard Settings. The reported scores denote the classification accuracy (%) and error bar (±) over 10 trials. We highlight the best score on each dataset in bold and the second score with underline.**

| Backbones | Methods | CiteSeer | | | PubMed | | | Cora | | |
|---|---|---|---|---|---|---|---|---|---|---|
| | | Full Test | High Hom Test | Low Hom Test | Full Test | High Hom Test | Low Hom Test | Full Test | High Hom Test | Low Hom Test |
| LINKX | ERM | 73.19 ± 0.99 | 73.89 ± 1.51 | 72.79 ± 1.47 | 87.86 ± 0.77 | 88.49 ± 1.37 | 87.19 ± 1.51 | 84.64 ± 1.13 | 85.13 ± 1.83 | 83.85 ± 1.99 |
| | ReNode | 73.25 ± 0.89 | 74.00 ± 1.21 | 72.89 ± 1.87 | 87.91 ± 0.72 | 88.58 ± 1.42 | 87.21 ± 1.71 | 84.70 ± 1.23 | 85.28 ± 1.99 | 84.24 ± 2.13 |
| | SRGNN | 73.27 ± 0.99 | 74.03 ± 1.11 | 72.85 ± 1.87 | 87.96 ± 0.81 | 88.68 ± 1.72 | 87.31 ± 1.51 | 84.71 ± 1.25 | 85.22 ± 1.87 | 84.34 ± 2.03 |
| | StruRW-Mixup | 73.29 ± 0.91 | 73.93 ± 1.25 | 72.99 ± 1.91 | 88.12 ± 0.51 | 88.71 ± 1.44 | 87.58 ± 1.59 | 84.67 ± 1.54 | 85.33 ± 1.91 | 84.34 ± 2.43 |
| | EERM | 73.17 ± 0.79 | 73.81 ± 1.45 | 73.09 ± 1.65 | 87.96 ± 0.84 | 88.59 ± 1.32 | 87.29 ± 1.63 | 84.62 ± 1.37 | 85.14 ± 1.63 | 83.87 ± 2.03 |
| | BAGNN | 73.33 ± 0.88 | 73.99 ± 1.61 | 73.19 ± 1.63 | 88.01 ± 0.94 | 88.78 ± 1.57 | 87.69 ± 1.39 | 84.60 ± 1.28 | 85.24 ± 1.83 | 83.84 ± 2.41 |
| | FLOOD | 73.34 ± 0.91 | 73.95 ± 1.55 | 73.22 ± 1.67 | 88.05 ± 0.95 | 88.84 ± 1.42 | 87.81 ± 1.59 | 84.72 ± 1.41 | 85.35 ± 1.63 | 83.99 ± 2.51 |
| | CaNet | 73.38 ± 0.95 | 74.08 ± 1.54 | 73.31 ± 1.55 | 88.11 ± 0.98 | 88.89 ± 1.67 | 87.91 ± 1.54 | 84.81 ± 1.31 | 85.39 ± 1.57 | 84.11 ± 2.58 |
| | IENE | 73.43 ± 0.97 | 74.15 ± 1.58 | 73.32 ± 1.87 | 88.12 ± 0.94 | 88.90 ± 1.65 | 87.90 ± 1.66 | 84.92 ± 1.45 | 85.41 ± 1.68 | 84.45 ± 2.81 |
| | GDN | 73.31 ± 0.81 | 73.99 ± 1.61 | 73.21 ± 1.68 | 88.14 ± 0.94 | 88.91 ± 1.64 | 87.92 ± 1.41 | 84.64 ± 1.33 | 85.27 ± 1.69 | 83.91 ± 2.78 |
| | **HEI(Ours)** | **73.51 ± 0.81** | **74.18 ± 1.25** | **73.42 ± 1.85** | **88.50 ± 0.97** | **89.01 ± 1.24** | **87.99 ± 1.92** | **85.17 ± 1.53** | **85.44 ± 1.83** | **84.84 ± 1.97** |
| GloGNN++ | ERM | 77.22 ± 1.78 | 78.15 ± 2.55 | 76.79 ± 2.54 | 89.24 ± 0.39 | 90.62 ± 0.99 | 88.75 ± 1.28 | 88.33 ± 1.09 | 90.06 ± 1.52 | 87.37 ± 1.64 |
| | ReNode | 77.31 ± 1.69 | 78.27 ± 2.48 | 76.90 ± 2.39 | 89.25 ± 0.35 | 90.64 ± 0.87 | 88.79 ± 1.24 | 88.39 ± 1.21 | 90.11 ± 1.49 | 87.45 ± 1.57 |
| | SRGNN | 77.33 ± 1.65 | 78.24 ± 2.75 | 76.91 ± 2.77 | 89.33 ± 0.51 | 90.81 ± 1.21 | 88.99 ± 1.57 | 88.53 ± 1.09 | 90.46 ± 1.53 | 87.58 ± 1.54 |
| | StruRW-Mixup | 77.35 ± 1.57 | 78.27 ± 2.11 | 76.90 ± 2.42 | 89.48 ± 0.44 | 90.81 ± 0.97 | 89.17 ± 1.33 | 88.39 ± 1.44 | 90.15 ± 1.69 | 87.85 ± 1.77 |
| | EERM | 77.35 ± 1.81 | 78.27 ± 2.45 | 76.89 ± 2.81 | 89.34 ± 0.39 | 90.82 ± 1.09 | 88.95 ± 1.38 | 88.39 ± 1.21 | 90.36 ± 1.42 | 87.47 ± 1.74 |
| | BAGNN | 77.42 ± 1.81 | 78.35 ± 2.81 | 76.89 ± 2.42 | 89.37 ± 0.45 | 90.87 ± 1.29 | 88.99 ± 1.58 | 88.49 ± 1.31 | 90.39 ± 1.47 | 87.81 ± 1.64 |
| | FLOOD | 77.43 ± 1.79 | 78.39 ± 2.51 | 76.95 ± 2.37 | 89.41 ± 0.51 | 90.91 ± 1.29 | 89.08 ± 1.58 | 88.51 ± 1.27 | 90.40 ± 1.51 | 87.88 ± 1.69 |
| | CaNet | 77.42 ± 1.81 | 78.32 ± 2.54 | 76.75 ± 2.87 | 89.51 ± 0.58 | 90.97 ± 1.29 | 89.19 ± 1.61 | 88.54 ± 1.33 | 90.51 ± 1.61 | 88.11 ± 1.58 |
| | IENE | 77.43 ± 1.79 | 78.39 ± 2.51 | 76.96 ± 2.58 | 89.52 ± 0.57 | 91.00 ± 1.44 | 89.22 ± 1.62 | 88.78 ± 1.37 | 90.79 ± 1.47 | 88.28 ± 1.49 |
| | GDN | 77.44 ± 1.51 | 78.65 ± 3.75 | 76.97 ± 2.53 | 89.39 ± 0.51 | 90.97 ± 1.01 | 88.99 ± 1.21 | 88.29 ± 1.34 | 90.44 ± 1.91 | 87.89 ± 1.22 |
| | **HEI(Ours)** | **77.85 ± 1.89** | **79.11 ± 2.59** | **77.30 ± 2.85** | **89.99 ± 0.39** | **91.52 ± 0.99** | **89.48 ± 1.33** | **88.93 ± 1.19** | **90.97 ± 1.39** | **88.47 ± 1.74** |

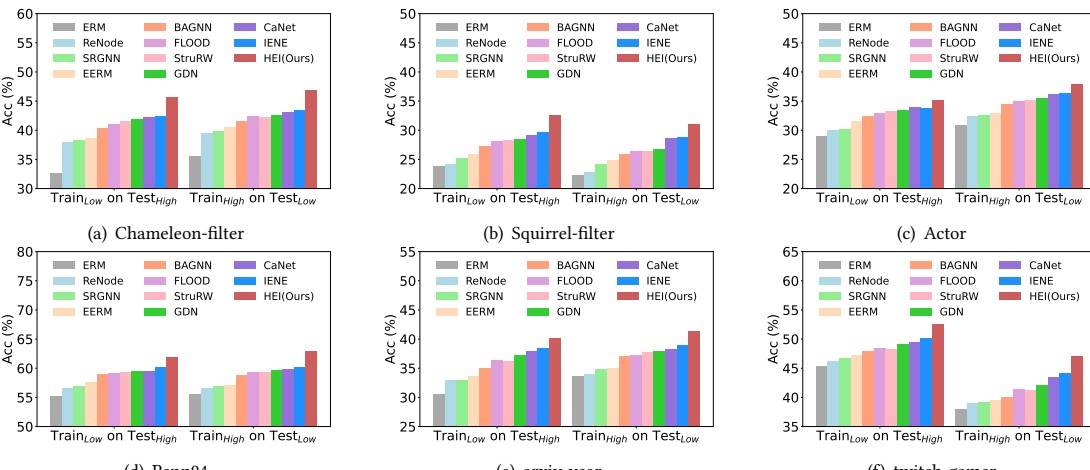

(a) Chameleon-filter      (b) Squirrel-filter      (c) Actor

(d) Penn94      (e) arxiv-year      (f) twitch-gamer

**Figure 6: Comparison experiments under Simulation Settings where exists severe distribution shift between train and test, including $Train_{High}$ on $Test_{Low}$ and $Train_{Low}$ on $Test_{High}$. We adopt the GloGNN++ as the backbone there.**

**Table 7: Comparison experiments on large-scale datasets when we respectively adopt local similarity(Local Sim), post-aggregation similarity (Agg Sim), and SimRank as indicators to estimate nodes' neighbor patterns so as to infer latent environments under Standard settings.**

| Backbones | Methods | Penn94 | | | arxiv-year | | | twitch-gamer | | |
|---|---|---|---|---|---|---|---|---|---|---|
| | | Full Test | High Hom Test | Low Hom Test | Full Test | High Hom Test | Low Hom Test | Full Test | High Hom Test | Low Hom Test |
| LINKX | HEI (Local Sim) | 85.12 ± 0.21 | 88.28 ± 0.33 | 82.15 ± 0.59 | 54.41 ± 0.21 | 64.23 ± 0.47 | 48.29 ± 0.22 | 66.18 ± 0.12 | 83.75 ± 0.34 | 48.12 ± 0.47 |
| | HEI (Agg Sim) | 85.21 ± 0.17 | 88.29 ± 0.38 | 82.22 ± 0.54 | 54.45 ± 0.23 | 64.33 ± 0.49 | 48.33 ± 0.32 | 66.21 ± 0.15 | 83..85 ± 0.39 | 48.45 ± 0.57 |
| | HEI (SimRank) | **86.22 ± 0.28** | **89.24 ± 0.28** | **83.22 ± 0.59** | **56.05 ± 0.22** | **66.53 ± 0.41** | **49.33 ± 0.32** | **66.79 ± 0.14** | **85.33 ± 0.25** | **49.21 ± 0.57** |
| GloGNN++ | HEI (Local Sim) | 86.08 ± 0.24 | 89.70 ± 0.64 | 82.18 ± 0.37 | 54.42 ± 0.24 | 64.48 ± 1.54 | 48.55 ± 0.64 | 66.30 ± 0.18 | 83.21 ± 0.68 | 49.00 ± 0.67 |
| | HEI (Agg Sim) | 86.15 ± 0.25 | 89.70 ± 0.69 | 82.43 ± 0.38 | 54.44 ± 0.25 | 64.51 ± 1.54 | 48.69 ± 0.81 | 66.34 ± 0.21 | 83.19 ± 0.78 | 49.14 ± 0.57 |
| | HEI (SimRank) | **87.18 ± 0.28** | **89.99 ± 0.65** | **83.59 ± 0.39** | **55.71 ± 0.24** | **66.29 ± 1.14** | **49.52 ± 0.75** | **66.99 ± 0.17** | **84.37 ± 0.68** | **50.40 ± 0.52** |

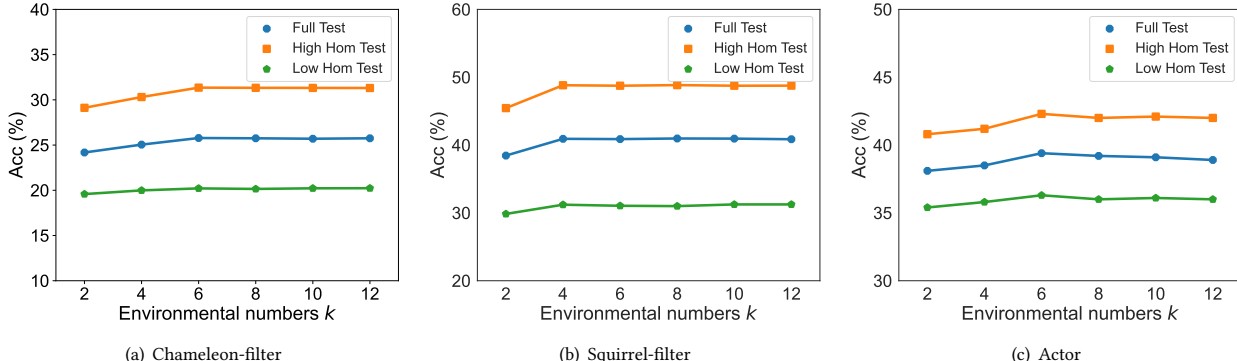

(a) Chameleon-filter

(b) Squirrel-filter

(c) Actor

Figure 7: Parameter Sensitivity of environmental numbers $k$ under Standard Settings.

utilizes the variance from multiple virtual environments as regularization to help model training. We entirely follow this work to address graph structure distribution shifts on heterophilic graphs.

**BAGNN**[9], Bias-aware(BA) GNN is also a method that deals with agnostic distribution shifts on graphs from the perspective of invariant learning. Specifically, it can be summarized into two steps: the environment clustering module assigns nodes to environments with minimization loss, and the invariant graph learning module learns invariant representation across environments with minimization loss. During the process of environment clustering, it adopts the masking strategy with graph augmentation. We entirely follow this work to address graph structure distribution shifts on heterophilic graphs.

**FLOOD**[24], it is also a a flexible invariant learning framework for OOD generalization on graphs. Specifically, it includes two key modules, invariant learning, and bootstrapped learning. The invariant learning modules construct multiple environments from graph data augmentation and learn invariant representation under risk extrapolation. Besides, the bootstrapped learning component is inspired by test time adaptation, which proposes to train a shared graph encoder with the invariant learning part according to the test distribution. We entirely follow this work to address graph structure distribution shifts on heterophilic graphs.

**CaNet**[36], it is a recently proposed invariant learning framework that integrates an environment estimator with a mixture-of-expert GNN predictor. , aiming to train robust GNNs under node-level distribution shifts. Exactly, it holds the findings that the crux of GNNs' failure in OOD generalization lies in the latent confounding bias from the environment and proposes to estimate the pseudo environments for each layer of the GNN network, assisted causal inference. Their defined environments are different from the environment we clarify in the paper, which is just stated from the perspective of feature separation. We entirely follow this work to address graph structure distribution shifts on heterophilic graphs.

**IENE**[39], it is also a recently proposed invariant learning framework that identifies and extrapolates the node environment for Out-of-Distribution Generalization on graphs. However, for extrapolating topological environments, they still adopt graph augmentation techniques to identify structural invariance, which is indeed different from our strategy for inferring environments. We entirely

follow this work to address graph structure distribution shifts on heterophilic graphs.

**StruRW**[23], it is a structure-reweighting method originally designed for a new type of conditional structure shift (CSS), which the current Graph domain adaptation approaches are provably suboptimal to deal with. We entirely follow this work to address graph structure distribution shifts on heterophilic graphs.

**GDN**[36], it is a prototype learning method originally designed for Graph Anomaly Detection. It teases out the anomaly features and mitigates the effect of heterophilic neighbors by devising a dynamically optimized prototype vector to guide the node representation learning under graph structure distribution shift. We entirely follow this work to address graph structure distribution shifts on heterophilic graphs.

**HEI (Ours)**, our training process can be concluded as follows: Given a heterophilic graph input, we first calculate the SimRank for each node in advance. Then, based on Eq. 9, we collectively learn environment partition and invariant representation on heterophilic graphs, assisted by SimRank, to address graph structure distribution shifts on heterophilic graphs. Therefore, we save the processed SimRank values on nodes on the graph in advance and transfer them into tensors for the training. For training details, we should warm up for some epochs to avoid the learned environments in the initial stage that are not effective, which may influence the optimization of models. So at the beginning warm-up stage, we adopt the ERM strategy. After that, we adopt our proposed framework to learn an invariance penalty to improve model performance. For the range of parameters, we first execute experiments using basic backbones to get the best parameters of num-layers and hidden channels on different datasets. Then, we fix the num-layers and hidden channels to adjust other parameters, penalty weight$\lambda$) from $\{1e-, 1e-2, 1e-1, 1, 10, 100\}$, learning rate from $\{1e-2, 5e-3, 1e-3, 5e-4, 1e-4\}$ and weight decay from $\{1e-2, 5e-3, 1e-3\}$. We also provide parameter sensitivity of environment number $k$ in the paper. Moreover, the $\rho$ is a two-layer MLP with the hidden channel from $\{16, 32, 64\}$, and its learning rate should be lower than the backbone in our experiments, within the range from $\{5e-3, 1e-3, 5e-4, 1e-4\}$.

