# OpenReview forum: "Leveraging Invariant Principle for Heterophilic Graph Structure Distribution Shifts"
_ACM.org/TheWebConf/2025/Conference — WWW 2025 Oral_

### Official Review · Reviewer_HaHw · 2024-11-18

**Novelty:** 6
**Technical Quality:** 5

**Review:**

Summary:

This paper addresses the impact of heterogeneous data distributions on downstream tasks and proposes HEI, a framework designed to generate invariant node representations by incorporating heterophily information and estimated neighbor patterns to infer latent environments without augmentation, which are then used for invariant prediction.

Strengths:

1. Strong Innovation: The paper introduces a novel perspective on existing issues in heterogeneous graph learning, particularly highlighting the limitations of current methods in handling different heterogeneous data distributions.

2. Comprehensive Experimental Validation: The authors provide a thorough comparison with existing work, validating their proposed method from multiple angles.

3. Good Writing: The writing is clear, detailed, and meets the expected standards.

Weaknesses:

1. Formatting Issues: The format for Figure 1(a3) should be Figure 1 (a3). Similar issues appear throughout the introduction.

2. Lack of Clarity in the Introduction: While the introduction provides detailed background on the problem, the description of the proposed method is vague: “HEI, a framework capable of generating invariant node representations through incorporating heterophily information to infer latent environments.” This statement is unclear and does not adequately explain the specific contributions or methodology of the paper.

3. Integration of the Proposed Method: In the introduction, it would be helpful to integrate the problem background with the proposed solution, explaining how the method directly addresses existing limitations. This would provide readers with a clearer understanding of the authors' contributions.

4. Quantification of Results in Section 4.2: In subsection 4.2, the authors should include more quantitative descriptions of the data in the tables rather than just summarizing the results.

5. Figure 3 Caption Formatting: The caption for Figure 3 has "Train" and "Test" in italics, which does not match the style in the figure.

6. Citation Inconsistencies: The formatting of journal and conference names in the references is inconsistent, with capitalization errors in some instances.

**Questions:**

Questions:

1. The term “environment” is somewhat vague in the paper. In the fourth paragraph of the introduction, what exactly does “explicit environments” refer to? In Figure 1, what does “Different environment” represent? Are these referring to different neighbor distributions?

2. In the dataset splits, what are the median homophily values for each of the datasets?

3. Does the proposed method have an advantage in terms of computational cost compared to the baselines? The proposed method seems to require processing for each node individually, which could be time-consuming.

**Reviewer Confidence:**

4: The reviewer is certain that the evaluation is correct and very familiar with the relevant literature

**Scope:**

3: The work is somewhat relevant to the Web and to the track, and is of narrow interest to a sub-community

---

### Official Review · Reviewer_rt81 · 2024-11-29

**Novelty:** 3
**Technical Quality:** 4

**Review:**

This paper introduces a novel approach to addressing heterophilic graph structure distribution shifts in semi-supervised learning tasks. By leveraging node feature similarity to estimate neighbor patterns, the authors propose a framework that learns invariant node representations without relying on data augmentation. The method demonstrates strong performance across diverse benchmarks and backbones compared to state-of-the-art baselines. Here are the pros and cons of the article.
Pros:
(1) The paper provides a clear and detailed description of the problem. In the Introduction section, the authors effectively utilize various graphical methods to illustrate the challenges posed by heterophilic graph structure distribution shifts and highlight the limitations of existing approaches.
(2) The proposed framework builds environment partitions and learns invariant node representations without data augmentation, which improves computational efficiency.
(3) The authors conduct extensive experiments on various benchmarks and backbones, demonstrating the effectiveness and robustness of their method.
Cons:
(1) The paper's description of the problem of OOD on heterophilic graphs is still vague, and a formal definition would have been better.
(2) The scope of out-of-distribution (OOD) problems addressed by the paper appears to be limited. Specifically, the proposed method focuses on addressing distribution shifts in heterophilic graphs by leveraging a measure of node homophily and designing a targeted solution. However, its applicability may be restricted in cases involving other types of distribution shifts, such as feature shifts, where the proposed approach may not perform as effectively.
(3) Some descriptions in the paper are somewhat redundant. For instance, Equation (5) attempts to demonstrate that node similarity can be leveraged as an indicator of neighbor patterns without relying on label information. However, clustering is inherently a method based on similarity for grouping or classification, making such a proof unnecessary and potentially redundant.

**Questions:**

(1) The paper appears to focus solely on experiments conducted in the transductive setting. Are there any experimental results or discussions regarding the method's performance in the inductive setting?
(2) The article dedicates a substantial section to explaining why similarity is used to measure neighbor patterns. However, the current explanation lacks clarity and depth, making it difficult to fully understand the rationale and its significance.

**Reviewer Confidence:**

3: The reviewer is confident but not certain that the evaluation is correct

**Scope:**

3: The work is somewhat relevant to the Web and to the track, and is of narrow interest to a sub-community

---

### Official Review · Reviewer_E7mo · 2024-12-02

**Novelty:** 4
**Technical Quality:** 5

**Review:**

Importance of the Problem
The paper focuses on the issue of Heterogeneous Graph Structure Shift (HGSS), a problem that has not been sufficiently explored in existing Graph Neural Network (GNN) research but is significant for handling complex real-world graph data.

Innovative Method
The authors propose the HEI (Heterophily-Guided Environment Inference) framework, which infers implicit environmental partitions by utilizing node neighbor pattern information without relying on traditional data augmentation, effectively addressing the distribution shift problem. This method is well-supported by both theoretical analysis and experimental evaluation, demonstrating a degree of innovation.

Experimental Validation
The experimental section compares various existing optimal methods (including state-of-the-art HGNN and methods based on invariance learning), showing that HEI outperforms existing methods on multiple heterogeneous and homogeneous graph datasets, particularly in scenarios with significant distribution differences.

Theoretical Support
The paper provides thorough theoretical analysis, explaining the rationale behind the HEI framework and demonstrating its advantages in addressing the HGSS problem.

**Questions:**

Insufficient Background Introduction
Although the paper mentions the shortcomings of existing methods, the background explanations for core concepts such as environmental inference and node pattern estimation are not detailed enough, which may lead to difficulties in understanding for readers.

Limited Experimental Scope
The experiments primarily focus on a few public datasets. While these datasets are representative, there is a lack of discussion on larger-scale practical application scenarios.

Insufficient Quantitative Comparison Analysis with Existing Methods
Although the paper compares various methods, the analysis of specific performance gaps and reasons for certain baseline methods (such as SRGNN and EERM) is relatively simplistic. A deeper exploration of why existing methods have failed to effectively address the HGSS problem would be beneficial.

Inadequate Parameter Sensitivity Analysis
While the paper studies the impact of the number of training environments on the model, it lacks comprehensive sensitivity analysis for other important hyperparameters (such as the choice of similarity computation methods and the weights for neighbor pattern estimation).

Insufficient Robustness and Generalization Testing
There is a lack of testing for adaptability in extreme scenarios (such as highly heterogeneous graphs and dynamic graphs); the experiments are solely based on static graphs and do not cover temporal variations in practical applications.

High Information Density in Graphs
The graphs in the experimental section (such as Tables 1, 2, and Figure 3) contain too much information, lacking intuitive summaries and analyses, making it difficult for readers to quickly grasp key conclusions.

**Reviewer Confidence:**

4: The reviewer is certain that the evaluation is correct and very familiar with the relevant literature

**Scope:**

4: The work is relevant to the Web and to the track, and is of broad interest to the community

---

### Official Review · Reviewer_UALA · 2024-12-02

**Novelty:** 6
**Technical Quality:** 6

**Review:**

This paper addresses the critical issue of heterophilic graph structure distribution shifts (HGSS), a challenge that has been overlooked in the design of heterophilic graph neural networks (HGNNs). The authors propose a novel framework called HEI (Heterophily-Guided Environment Inference), which innovatively leverages node neighbor patterns to infer latent environments and generate invariant node representations without relying on data augmentation. Extensive theoretical analysis and experiments demonstrate the framework's effectiveness across a wide range of benchmarks, including both heterophilic and homophilic graph datasets. HEI outperforms state-of-the-art methods in handling distribution shifts and generalizing under various challenging scenarios.

**Questions:**

Pros:

- The paper identifies and formalizes the overlooked issue of distribution shifts in heterophilic graphs, which is significant for the graph learning community,
- The authors provide detailed theoretical guarantees and causal analysis, grounding the framework's design in strong foundations.
- HEI achieves significant improvements over state-of-the-art methods across diverse datasets, particularly in settings with severe distribution shifts.
- The computational complexity analysis confirms the scalability, making it suitable for large-scale graphs. Notably, the authors have included experiments on large-scale datasets, which is encouraged.

Cons:

- My main question is about the similarity-based mechanism. Does this imply that the authors have made some assumptions, such as a correlation between the similarity of node features and the level of homophily? Does the effectiveness of HEI depend highly on the choice of similarity metric? The authors are recommended to discuss this in detail.
- Could the authors provide implementation details, such as the source code?

**Reviewer Confidence:**

4: The reviewer is certain that the evaluation is correct and very familiar with the relevant literature

**Scope:**

4: The work is relevant to the Web and to the track, and is of broad interest to the community

---

### Official Review · Reviewer_3cCB · 2024-12-03

**Novelty:** 4
**Technical Quality:** 3

**Review:**

This paper tackles the challenging problem of handling data distribution shifts in heterophilic graph neural networks by learning invariant node representations. The authors propose HEI, a novel framework that incorporates heterophily information and estimated neighbor patterns to infer latent environments without requiring data augmentation. These inferred environments are then utilized to achieve invariant predictions.  The framework is well-motivated, addressing the structural differences and distribution shifts inherent in heterophilic graphs, which many existing methods struggle with. By leveraging heterophily information, the proposed method provides a principled way to model the unique challenges posed by these graphs.  The authors validate their approach with extensive experiments across various benchmark datasets and backbone models. The results demonstrate the effectiveness and robustness of HEI compared to state-of-the-art baselines. This suggests the method's potential for broader applicability and reliability in real-world scenarios where heterophily is prevalent.

**Questions:**

The research presented in this paper falls outside the scope of my expertise. As a result, my score is based on an estimation and may not accurately reflect the quality of the paper.

**Reviewer Confidence:**

1: The reviewer's evaluation is an educated guess

**Scope:**

4: The work is relevant to the Web and to the track, and is of broad interest to the community